# Gut Protective Effect from Newly Isolated Bacteria as Probiotics against Dextran Sulfate Sodium and Carrageenan-Induced Ulcerative Colitis

**DOI:** 10.3390/microorganisms11071858

**Published:** 2023-07-23

**Authors:** Yuka Ikeda, Ai Tsuji, Satoru Matsuda

**Affiliations:** Department of Food Science and Nutrition, Nara Women’s University, Kita-Uoya Nishimachi, Nara 630-8506, Japan

**Keywords:** ulcerative colitis, inflammatory bowel disease, gut microbiota, probiotics, PI3K, AKT, dextran sodium sulfate, carrageenan

## Abstract

Gut microbiome dysbiosis might be linked to certain diseases such as inflammatory bowel diseases (IBDs), which are categorized by vigorous inflammation of the gastrointestinal tract. Several studies have shown the favorable anti-inflammatory effect of certain probiotics in IBD therapy. In the present investigation, the possible gut protective effects of commensal bacteria were examined in an IBD model mouse that was cost-effectively induced with low molecular weight dextran sulfate sodium (DSS) and kappa carrageenan. Our conclusions show that certain probiotic supplementation could result in the attenuation of the disease condition in the IBD mouse, suggesting a favorable therapeutic capability for considerably improving symptoms of gut inflammation with an impact on the IBD therapy. However, the molecular mechanisms require further investigation.

## 1. Introduction

Inflammatory bowel diseases (IBDs) including ulcerative colitis (UC) and Crohn’s disease are a kind of chronic inflammatory disorders of the colon and/or intestine. The morbidity of IBD has been increasing worldwide [1]. UC is untiring and limited to the colon and rectum, whereas the lesions of Crohn’s disease are sporadic and comprise all parts of the gastrointestinal tract [2]. UC is considered by the ulcer formation in mucosa and the infiltration of neutrophils and/or lymphocytes into the membrane space of the mucosa [3]. The clinical appearances of UC are diarrhea, abdominal pain, severe bloody stools, and so forth [4]. Although the etiology of IBD remains unidentified, imbalance of the inflammatory responses, damage to the mucosal barrier, and alterations of the gut microbiota might be considered as risk factors [5]. Environmental factors and/or genetic reasons might be involved in the pathogenesis. However, the precise mechanisms that trigger IBD remain poorly understood [6]. Although immunosuppressive and/or anti-oxidative chemicals have been used in the treatment of IBD, these treatments may have shown long-term toxicity with unfavorable side-effects [7]. For example, many novel therapeutic agents such as adalimumab or tacrolimus have become available to treat UC [8], however, they increase the risk of severe infection in the host because of their immune-suppression [9]. Accordingly, it is imperative to explore safe treatments such as dietary interventions for the option of IBD remedies.

Comparable symptoms could be induced in mice with the addition of dextran sulfate sodium (DSS) for the UC model. With this UC mouse model, several investigations have also proposed that IBD might be strongly linked to the intestinal microenvironment of the patient, inappropriate immune response, and the intestinal microbiome [10]. IBD may be reflected to result from an inappropriate immune response to the gut microbiota [11], which could induce tissue damage of the intestine, and perturbation of the epithelial barrier function as well as the gut microenvironment. The perturbation might maintain a relationship between the dysbiosis of the gut microbiota and the pathology of IBD. Gut microbial dysbiosis may also result from alterations in the commensal microbiota composition [12]. Consequently, the gut microbiota might be an encouraging therapeutic target for IBD. It has been revealed that certain probiotics could work as a good mediator of anti-inflammation in IBD pathology to repair the composition of the gut microbiota [13]. Correspondingly, the symptoms of UC have been dampened by the administration of probiotic bacteria, which might also alleviate the barrier damage in the gut epithelium [14]. Furthermore, various studies have shown that the beneficial action of probiotics could improve the condition in IBD via the modification of the gut microbiota composition [15]. However, the molecular mechanisms of probiotics with health-advantageous effects remain precisely unidentified. In addition to probiotics being potential inhibitors of pathogenic microorganism growth, certain probiotics can produce short-chain fatty acids (SCFAs) that are important to maintain the gut mucosal integrity. SCFAs may also possess potential anti-inflammatory and/or anti-oxidative functions [16]. Administration of *Lactobacillus* and/or *Bifidobacteria* strains as adjuvant therapy could significantly improve the clinical symptoms as well as the disease progression of UC patients [17]. Although the fundamental mechanisms have not been well-clarified yet, the phosphatidylinositol 3-kinase (PI3K)/protein kinase B (AKT) signaling pathway seems to be involved in the pathogenesis of IBD, which might also be closely related to the inflammatory responses and/or oxidative stress [18]. Furthermore, several treatments with certain probiotics and/or the combination of probiotics with traditional treatments for UC have increasingly gained attention [19]. 

The gut microbiota could play an imperative role in human health as well as in various inflammatory diseases. While *Bacteroides* and/or *Pseudomonas* could induce colitis [20], certain bacteria might encourage the establishment of good microbiota in the gut that may help to reduce the development of colitis. For this purpose, it has been challenging to identify the specific bacteria that can suppress the oxidative stress and/or inflammation in UC. Recently, we have fortunately isolated several bacteria with high antioxidant capability from several fruits and/or vegetable materials (unpublished data, Bact A and Bact B derived from strawberry and banana, respectively). Here, we evaluated the effects of two clones of these bacteria on the development of DSS and the carrageenan-induced experimental colitis of mice. 

## 2. Materials and Methods

### 2.1. Materials

Dextran sulfate sodium (DSS, MW5000), fructose, and 10% formalin neutral buffer solution were obtained from Fujifilm Wako Pure Chemical Corporation (Osaka, Japan). κ-Carrageenan (CGN) was obtained from Tokyo Chemical Industry Co., Ltd. (Tokyo, Japan). DSS CGN, and fructose were dissolved in sterile water for the mice to drink.

### 2.2. Mice

Male Crl: ICR mice (5-week-old) were obtained from Charles River Laboratories Japan, Inc. (Kanagawa, Japan). The mice were housed in an environmentally controlled room, at approximately 20 °C and 60% humidity with a 12-h light/dark cycle (lights on at 07:00 and off at 19:00). One mouse was housed per cage. The care and treatment of the experimental animals conformed with the guidelines for the ethical treatment of laboratory animals established by Nara Women’s University (Nara, Japan) (Approval No. 21-01).

[Exp. 1]. After one week of acclimation, the mice were randomly divided into four groups; Ct, UC, UC/Bact A, and UC/Bact B (n = 5/group). UC was induced by the administration of UC water containing 1.5% (*w*/*v*) DSS, 0.5% (*w*/*v*) κ-CGN, and 1% (*w*/*v*) fructose. The UC group mice drank the UC water for 13 days. The UC/Bact A and UC/Bact B groups drank the UC water with bacteria isolated from the fruits, respectively, for 13 days. Both bacterial concentrations for the peroral administration were approximately 1 × 10^6^ CFU mL^−1^. The control group drank 1% fructose water. Afterward, all mice were sacrificed. We collected plasma samples and colon and spleen tissue. The colon length and spleen weights were measured. The colon was washed with saline solution to remove stool. Colons near the rectum were fixed with 10% formalin neutral buffer solution. The rest of the colons were stored at −80 °C until analysis. The schedule is shown in Figure 1A.

[Exp. 2]. After one week of acclimation, the mice were randomly divided into four groups; Ct, UC, UC/preBact A, and UC/preBact B (n = 5/group). In the first week, the Ct and UC group mice drank 1% fructose water. The UC/preBact A and UC/preBact B groups drank 1% fructose water with bacteria, which were the same ones in Exp. 1. From day 7, the UC, UC/preBact A, and UC/preBact B groups drank UC water, which was the same as those in Exp. 1. The Ct group drank 1% fructose water. On day 21, all mice were sacrificed. We collected plasma samples, colon, and spleen tissues as in Exp. 1. The schedule is shown in Figure 1B.

### 2.3. Disease Activity Index Score (DAI)

In the UC model animals, stool consistency, occult blood, and body weight loss were observed. All mice were monitored for these clinical manifestations and scored with reference to the disease activity index score (DAI) reported by Hu et al. (Table 1).

### 2.4. Histological Examination

The colon were fixed in 10% neutral buffered formalin solution. Paraffin embedding, fine slice, and H.E. (hematoxylin and eosin) staining were carried out at Genostaff, Inc. (Tokyo). Histopathological preparation was observed by an OLYMPUS digital camera (CAMEDIA C-5060) with a microscope (OLYMPUS CKX41). The absolute magnification was ×200. The epithelium loss, crypt damage, and infiltration of inflammatory cells were observed and scored with reference to the histological score reported by Kim et al. (Table 2).

### 2.5. Western Blotting

The colon (0.1 mg) was homogenized with 0.3 mL RIPA buffer. The homogenates were centrifuged at 15000 × g for 10 min at 4 °C to obtain the supernatants. The supernatants were mixed with the sodium dodecyl sulfate sample buffer. The mixtures were boiled for 3 min and analyzed by Western blotting. Translated membranes were immunostained with primary antibodies Anti-PI3K (BD Biosciences), Anti-AKT (GeneTex), Anti-phosphorylated AKT (GeneTex), Anti-IRAK (BD Biosciences), and Anti-actin. Anti-rabbit IgG or anti-mouse IgG horseradish peroxidase-conjugated was used as the second antibody. Immunoreactive bands were visualized using a Luminescence Reagent Set (Wako Pure Chemical Industries, Ltd.) and detected with an Image Quant LAS500 (GE Healthcare Japan Com., Tokyo, Japan). The intensities of the detected bands were calculated using ImageJ software v.1.8.0.

### 2.6. Statistical Analyses

Animal data were expressed as the mean ± standard error (SE). Differences in body weight gain among groups were analyzed by two-way analysis of variance (ANOVA). Other data were analyzed by one-way ANOVA and Dunnett tests. *P* < 0.05 was considered as a statistically significant difference. All statistical analyses were performed using GraphPad Prism version 5.0 (GraphPad Software, Inc., San Diego, CA, USA).

## 3. Results 

### 3.1. Characteristics and Disease Activity Index of UC Model Mice

We employed the arrangement of low molecular weight DSS and the kappa carrageenan mixture to make the UC group mouse model. The DSS/carrageenan concentrations and duration used in the present experiments for the development of UC were determined from the consequence of preliminary experiments and the related literature. The experimental design of all experiments is shown in Figure 1. As body weight might be a main characteristic of IBD, we first detailed the changes in body weight to examine the advantage of probiotics with Bact A or Bact B on weight loss or weight gain in Exp. 1 (Figure 1A and Figure 2A). In the present experimental condition, however, there was no significant difference in body weight alterations among groups (Figure 2A). In addition, no mice died during the entire experiment. After about a week in this experiment, severe rectal bleeding and diarrhea were particularly observed in the UC group mice. Subsequently, the symptoms of stool as well as body weight loss were scored and are shown in Figure 2B. Blood stool, diarrhea, and bleeding from the anus periphery were overall observed in the UC group in the last stage of the experiment. Accordingly, the UC group mice showed the greatest score in the disease activity index (DAI) (*p* < 0.05 vs. Ct), and the DAI score of the UC/Bact B group was higher than that in the Ct group (Figure 2B). On the other hand, the DAI score was lower in the mice drinking the Bact A bacteria than that in the UC group, which possibly shows that the UC mice without the Bact A treatments could trigger a significant disease activity. In particular, Bact A had considerably suppressed the symptoms of UC (Figure 2B). Shortening of the colon length and increased spleen weight may be the inflammation marker of the UC model mice [21,22]. However, the colon length was almost the same among the four groups (Figure 2C). In contrast, the spleen weight was increased in the UC group than that in the Ct group (Figure 2D), which was decreased with the treatments of Bact A or Bact B by −15% or −19% compared to the UC group, respectively (Figure 2D). 

### 3.2. Histological Result from the Colon

Afterward, we examined the effect of Bact A or Bact B on the histological damage of the colon. UC may be considered by histological findings such as the infiltration of inflammatory cells into the mucosa and submucosa, edema, mucosal thickening, and the destruction of epithelial cells. The pathology of this UC may also exhibit a disturbance of intestinal function with narrowing of the gut canal and thickened muscular layers in the colon. Reliably, our microscopic analysis also found that colon epithelial cells in the present UC mice were severely damaged (Figure 3A). In particular, the infiltration of inflammatory cells into the submucosa was obviously observed at the colon epithelium in the UC group. Furthermore, edematous tissue findings were also detected in the submucosa of the UC and/or UC/Bact B groups compared to the Ct group. While the crypts of the intestine in the Ct group were arranged orderly in a row and spaced closely, those of the UC group were arranged messily with rather wide spaces. Furthermore, the intestinal tract of the UC group mice was tapered, and the muscular layer of the UC group was extremely thickened, showing that intestinal epithelial cells in the UC group were harshly damaged (Figure 3A). Consequently, a mice model of IBD could be cost-effectively constructed with the combinational use of low molecular weight DSS and kappa carrageenan, as shown here. It has been observed that probiotics with the use of Bact A or Bact B could possess protective effects on gut injury in this UC model of mice (Figure 3A). Consistently, histological scores were also demonstrated (Figure 3B), indicating the comparable result of DAI score, as shown in Figure 2B. 

### 3.3. Examinations for Protein Expression in the UC Mice Colon 

The PI3K/AKT signaling pathway is considered as one of the major pathways for cellular/organ protection and/or cell survival. The PI3K protein expression of the UC and the UC/Bact B groups was higher than that of the Ct group, but the UC/Bact A group showed similar levels of PI3K protein expression to the Ct group (Figure 4A,B). Additionally, expressions of phospho-AKT protein were similar to the result of the PI3K protein expressions (Figure 4C,D). Similar to these proteins, the protein expression of IRAK in the UC was significantly higher than that in the Ct group and/or the UC/Bact A group (Figure 4E,F).

### 3.4. Potential Prevention of UC with the Use of Probiotics

In Exp. 2 (Figure 1B), some mice were pretreated with Bact A or Bact B for a week before conducting the UC pathology with a combination of low molecular weight DSS and kappa carrageenan. In this experiment, we tried to evaluate the capability of Bact A or Bact B for the prevention of UC development. Body weight gains were at almost similar levels in all subgroups (Figure 5A). As shown in Figure 5B, the DAI score again increased in the UC group than that in the Ct group. However, mice in the UC/preBact A and UC/preBact B groups were observed for mild clinical appearance in bloody stool and/or stool looseness compared to the UC (Figure 5B). The alterations in colon length and spleen weight in Exp. 2 were almost similar to those in Exp. 1 (Figure 5C,D). Furthermore, the ulceration was observed in the UC group by using H&E staining, which appeared to be less damaged in the UC/preBact A group (Figure 6A). Consistent histological scores were also shown (Figure 6B). In addition, the results of the protein analyses for the colon of the UC mice were almost similar to those in Exp. 1 (Figure 7A–D). Overall, the efficacy of improvement in the potential effect of preBact A or preBact B in Exp. 2 seems to be somewhat smaller compared to that of Bact A or Bact B in Exp. 1.

## 4. Discussion

It has been suggested that the use of DSS and carrageenan might cause ROS accumulation and inflammatory response on the gut epithelial cells to yield mucosal damage [23,24]. In the present study, we definitely observed considerable protective effects from Bact A or Bact B on the gut damage of the UC group mice with the use of DSS and carrageenan. Although there was no alteration in the colon size between the Ct group and UC group, substantial damage might have occurred in the gut with the use of low molecular weight DSS and kappa carrageenan. Both bacteria more or less relieved the severity of UC symptoms (Figure 2B). We preliminarily explored the viability of all of the used bacteria in the circumstances of high bile salts and/or strong acids in vitro. A survival rate of more than 70% appeared to be detected in the LB broth with acids (pH 2) and 80% in the LB broth containing 0.3% bile salt (unpublished data, and personal communication). These results support the colonization of these bacteria in the gut environment and vigorous survival. Multiple factors such as environmental changes in the gut and/or the alteration in the gut microbiota might be associated with the pathological condition of UC [25]. Disruption of the immune response in mucosa may produce surplus amounts of various inflammatory cytokines and/or matrix metalloproteinases, which could also increase oxidative stress, resulting in severe damage to the gut/colon [26]. Furthermore, the disruption of the gut epithelium might be linked to the elevated lipopolysaccharide (LPS) levels [27]. For example, it has been reported that chemotherapy-associated gastrointestinal toxicity might be related to the abundance of the LPS-producing bacteria [28]. It is plausible that both Bact A and Bact B could disturb the growth of such harmful bacteria in the gut.

The PI3K/AKT signaling pathway has been predicted as a potential target for the treatment of UC [29]. In addition, probiotics therapy could exert the anti-inflammatory activity in DSS-induced colitis by the modulation of the PI3K/AKT pathway [30], which could be activated through their phosphorylation [31]. It has been shown that a classical traditional Chinese medicinal formula could ameliorate DSS-induced colitis through the regulation of gut microbiota and the PI3K/AKT signaling pathway [32]. The host gut microbiota might be greatly affected with the use of kappa carrageenan, which could cause UC by altering the gut microbiota, significantly reducing the amount of *Akkermansia muciniphila*, a potent anti-inflammatory bacterium in the gut [33]. Since *Akkermansia muciniphila* acts on the PI3K/AKT signaling pathway [33], it is possible that Bact A, in particular, might control the quantity of *Akkermansia muciniphila*. The colon histological findings also suggest that certain bacteria could directly influence the colonic epithelium, which could prevent gut microvilli damage triggered by the treatment of DSS and carrageenan. Protein expression analyses in the colon may also suggest that certain bacteria could work to alleviate UC symptoms via the modulation of the PI3K/AKT and/or interleukin-1 receptor-associated kinase (IRAK) signaling pathways. The IRAK is the main kinase of the toll-like receptor (TLR)-mediated signaling pathway, which has been considered as a new target for treating several inflammatory diseases [34]. 

Bact A, Bact B, and some fecal microbiomes of the mice treated have been identified by 16S rRNA gene sequencing, which cannot be published here due to patents pending. However, the possible beneficial effects of these bacteria may be related to the anti-inflammatory and/or antioxidative effects [35]. In fact, it has been shown that probiotics with *Lactobacillus* strains could have anti-inflammatory and/or antioxidative effects [36]. In addition, *Bifidobacterium* species might also have these anti-inflammatory and/or antioxidative effects against IBD [37]. Likewise, *Clostridium butyricum (C. butyricum)* could also improve inflammation in the colon by decreasing the intensities of inflammatory cytokines [38,39]. *C. butyricum* could increase the production of butyric acid, which may protect the epithelia through decreasing oxidative damage in the gut epithelium as well as in other organs [40,41,42]. In addition, butyrate could reduce the levels of microbiota-dependent endotoxins such as LPS [43]. In these ways, probiotics and/or prebiotics appear to be considered as effective protectants for several host organs, which might also improve the pathology of IBD. However, both Bact A and Bact B have been identified to neither *Lactobacillus*/*Bifidobacterium* species nor butyric acid producing bacteria strains. The molecular mechanisms by which Bact A or Bact B could employ beneficial effects against these UC model mice through the modification of the PI3K/AKT or IRAK signaling pathway in the colon are now under investigation and should be further explored in the future.

In addition, several limitations to this study should be described. First of all, the small sample size of the experiments may lead to non-significant results. Therefore, the results of the analyses in a small sample size study should be understood with additional caution. In particular, we cannot rule out the probability that the results of this study overstated the correctness due to the small number of animals used in each subgroup. Moreover, as the results of this study were attained by means of a mouse model, the transfer of our findings to humans should be thoughtfully assessed. Forthcoming investigations using a larger number of animals to address the above issues would be more informative.

## 5. Conclusions

A mice model of UC was built with the use of the DSS and carrageenan mixture. Several results in this study explain the potential mechanisms of certain bacteria including Bact A or Bact B in the prevention and/or inhibition of UC development, which may indicate them as good candidates for therapeutic options against UC and IBD. In addition, our study also revealed that the administration of certain probiotics may have a latent protective effect on the development of UC and IBD, which could be of great potential for the therapy of inflammatory diseases. Mechanistically, the protective properties might be in part related to the regulation of the PI3K/AKT or IRAK signaling pathways, which might be associated with anti-inflammatory and/or anti-oxidative effects. In conclusion, we found that both Bact A or Bact B bacteria could more or less improve the pathology of UC in the present animal model.

## Figures and Tables

**Figure 1 microorganisms-11-01858-f001:**
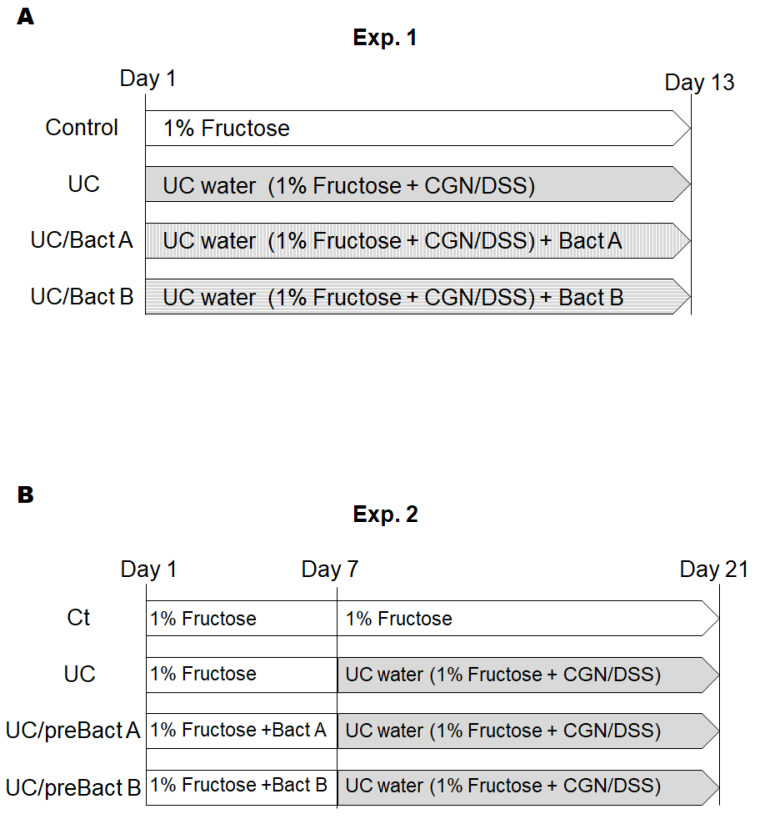
Study design. Study design of experiment 1 (**A**) and experiment 2 (**B**). Male ICR mice (5 weeks old) were divided into four groups in each experiment. In experiment 1, mice were sacrificed on day 13. In experiment 2, the mice changed water on day 7 and were sacrificed on day 21. CGN; carrageenan, DSS; dextran sodium sulfate.

**Figure 2 microorganisms-11-01858-f002:**
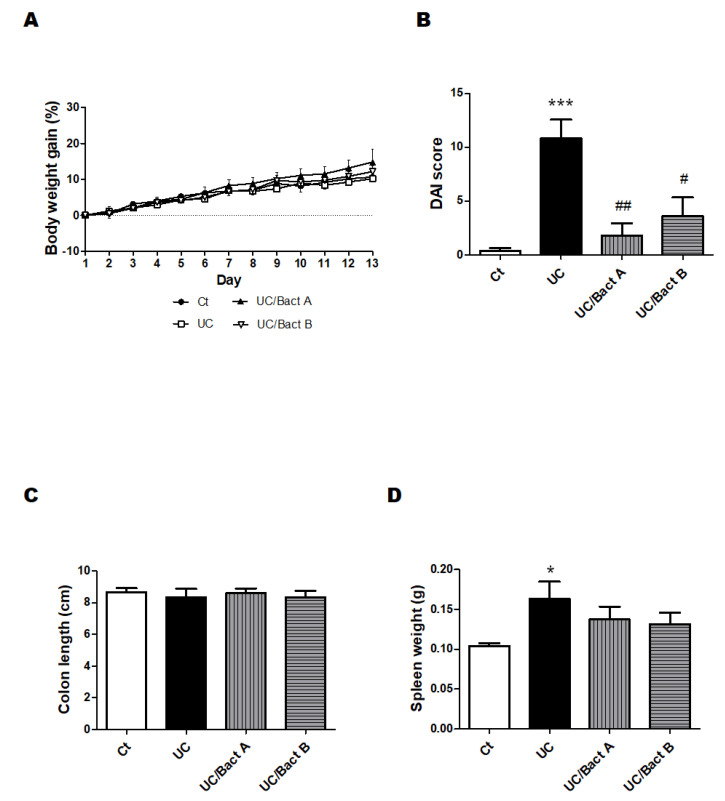
UC-induced mice showed clinical signs in Exp. 1. Some clinical signs of UC-induced mice and the effect of bacteria. (**A**) Body weights were quantified every day throughout the experiment. Ct group (black circle), UC group (white square), UC/Bact A group (black triangle), and UC/Bact B group (white inverted triangle); values are shown as the mean ± SE, n = 5/group. The data were verified by two-way ANOVA. (**B**) Disease activity index means the score that sums the daily score of five grades of weight loss, stool consistency, and four grades of occult blood levels. Ct group (white), UC group (black), UC/Bact A group (vertical stripes), and UC/Bact B group (horizontal stripes). Values are shown as the mean ± SE, n = 5/group. The data were verified by one-way ANOVA. (*** *p* < 0.005 vs. Ct group, ^#^
*p* < 0.05, ^##^
*p* < 0.01 vs. UC group) (**C**) Colon-size was gauged when the mice were dissected. Ct group (white), UC group (black), UC/Bact A group (vertical stripes), and UC/Bact B group (horizontal stripes). Values are shown as the mean ± SE, n = 5/group. The data were verified by one-way ANOVA. (**D**) Spleen weights were gaged when mice were dissected. Ct group (white), UC group (black), UC/Bact A group (vertical stripes), and UC/Bact B group (horizontal stripes). Values are shown as the mean ± SE, n = 5/group. The data were verified by one-way ANOVA. (* *p* < 0.05 vs. Ct group).

**Figure 3 microorganisms-11-01858-f003:**
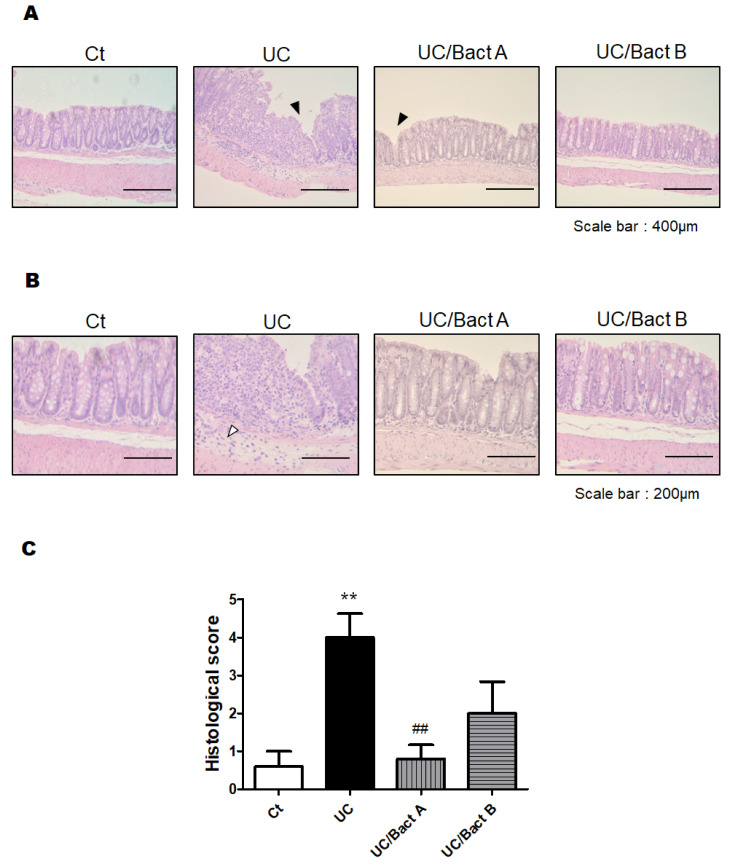
Histopathological examination of the colon in Exp. 1. (**A**) Standard H&E staining of the mice colon tissues. The final magnification was ×200. Each scale bar shows 400 µm. Black arrow shows the damage of the mucosa. (**B**) Standard H&E staining of the mice colon tissues. The final magnification was ×400. Each scale bar shows 200 µm. White arrow shows the infiltration of lymphocytes. (**C**) Histological scores were summed as the score of epithelium loss, crypt damage, and the infiltration of inflammatory cells. Ct group (white), UC group (black), UC/Bact A group (vertical stripes), and UC/Bact B group (horizontal stripes). Values are shown as the mean ± SE, n = 5/group. The data were verified by one-way ANOVA. (** *p* < 0.01 vs. Ct group, ^##^
*p* < 0.01 vs. UC group).

**Figure 4 microorganisms-11-01858-f004:**
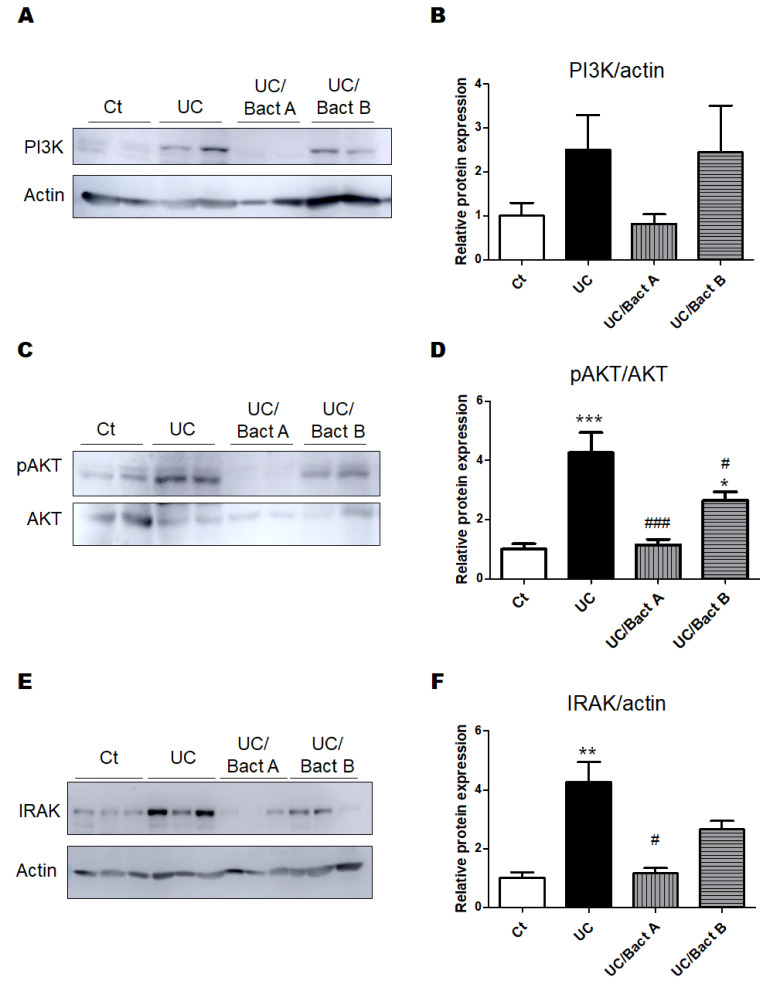
The protein expression in the colon in Exp. 1. (**A**) The image of PI3K and actin expression. (**B**) The protein expression of PI3K (85kDa) was quantified and normalized to that of actin by Western blot. Ct group (white), UC group (black), UC/Bact A group (vertical stripes), and UC/Bact B group (horizontal stripes). Values are shown as the mean ± SE, n = 5/group. The data were verified by one-way ANOVA. PI3K; phosphatidylinositol-3 kinase (**C**) The image of phosphor-AKT and AKT expression. (**D**) The protein expression of phosphor-AKT was quantified and normalized to that of AKT by Western blot. Ct group (white), UC group (black), UC/Bact A group (vertical stripes), and UC/Bact B group (horizontal stripes). Values are shown as the mean ± SE, n = 5/group. The data were verified by one-way ANOVA. (* *p* < 0.05, *** *p* < 0.005 vs. Ct group, ^#^
*p* < 0.05, ^###^
*p* < 0.005 vs. UC group) [E] The image of IRAK and actin expression. [F] The protein expression of IRAK was quantified and normalized to that of actin by Western blot. Ct group (white), UC group (black), UC/Bact A group (vertical stripes), and UC/Bact B group (horizontal stripes). Values are shown as the mean ± SE, n = 5/group. The data were verified by one-way ANOVA. (** *p* < 0.01 vs. Ct group, ^#^
*p* < 0.05 vs. UC group). IRAK, interleukin-1 receptor-associated kinase.

**Figure 5 microorganisms-11-01858-f005:**
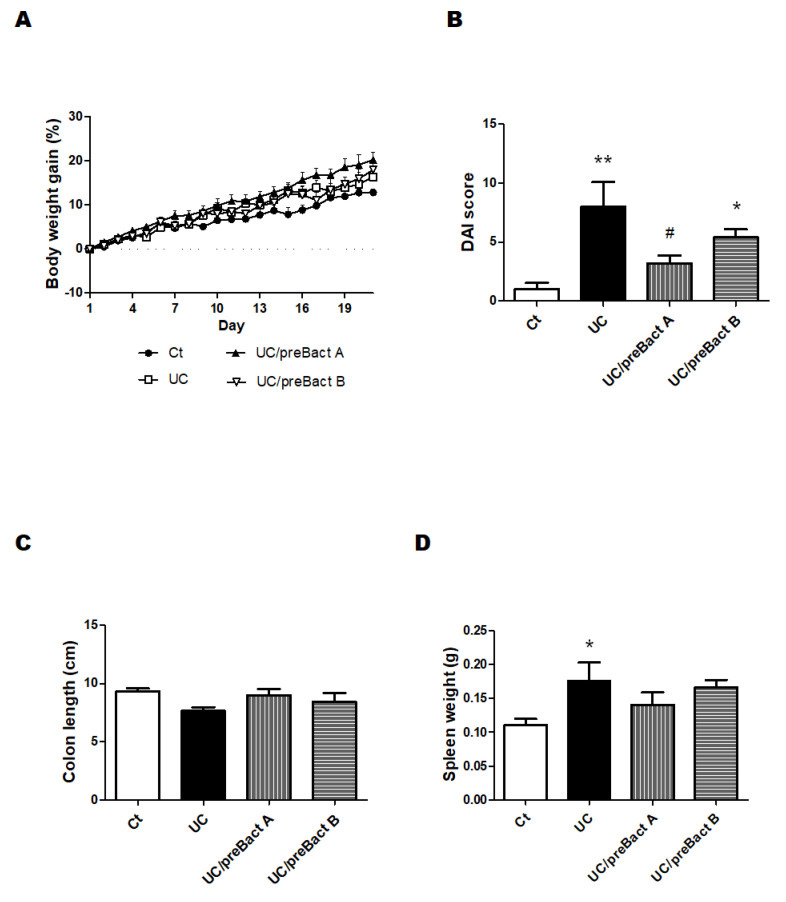
Clinical signs of UC-induced mice in the preventive treatment with Bact A and Bact B. Some clinical signs of UC-induced mice and the effect of bacteria. (**A**) Body weights were gauged every day throughout the experiment. Ct group (black circle), UC group (white square), UC/preBact A group (black triangle), and UC/preBact B group (white inverted triangle); values are shown as the mean ± SE, n = 5/group. The data were verified by two-way ANOVA. (**B**) Disease activity index is the score that sums the daily score of five grades of weight loss and stool consistency and four grades of occult blood. Ct group (white), UC group (black), UC/preBact A group (vertical stripes), and UC/preBact B group (horizontal stripes). Values are shown as the mean ± SE, n = 5/group. The data were verified by one-way ANOVA. (* *p* < 0.05, ** *p* < 0.01 vs. Ct group, ^#^
*p* < 0.05 vs. UC group) (**C**) Colon size was measured when the mice were dissected. Ct group (white), UC group (black), UC/preBact A group (vertical stripes), and UC/preBact B group (horizontal stripes). Values are shown as the mean ± SE, n = 5/group. The data were verified by one-way ANOVA. (**D**) Spleen weights were quantified when the mice were dissected. Ct group (white), UC group (black), UC/preBact A group (vertical stripes), and UC/preBact B group (horizontal stripes). Values are shown as the mean ± SE, n = 5/group. The data were verified by one-way ANOVA. (* *p* < 0.05 vs. Ct group).

**Figure 6 microorganisms-11-01858-f006:**
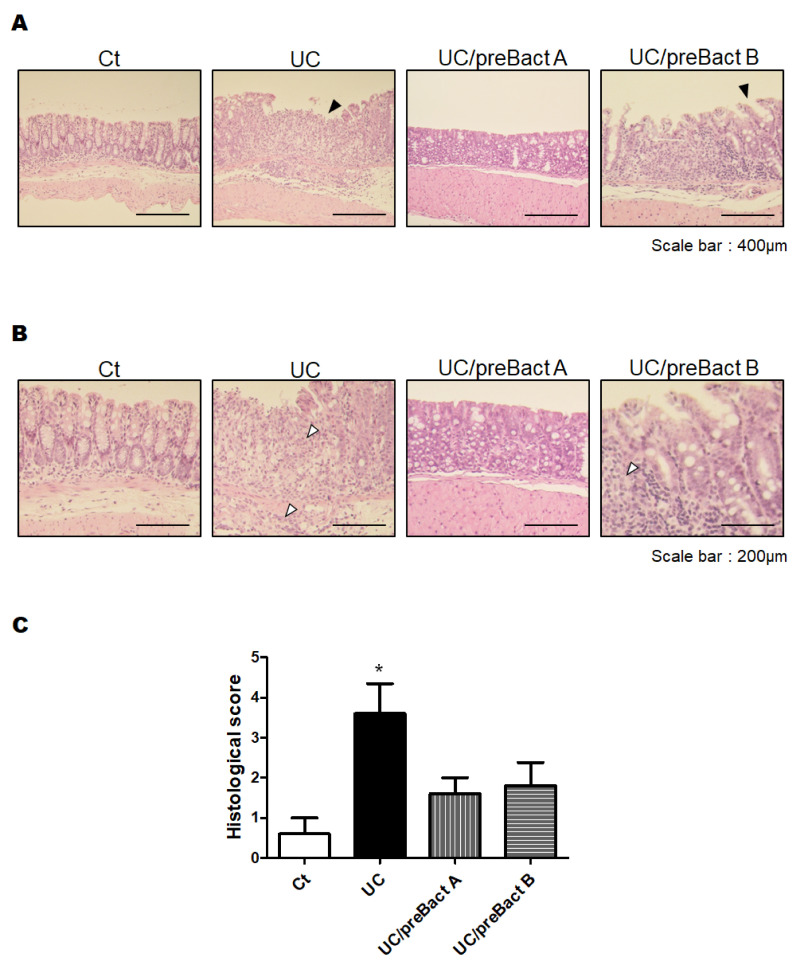
Histopathological analysis of the colon in the preventive treatment mice with Bact A and Bact B. (**A**) Standard H&E staining of the mice colon tissues. The final magnification was ×200. Each scale bar shows 400 µm. Black arrow shows the damage of the mucosa. (**B**) Standard H&E staining of the mice colon tissues. The final magnification was ×400. Each scale bar shows 200 µm. White arrow shows the infiltration of lymphocytes. (**C**) Histological score were summed as the score of epithelium loss, crypt damage, and the infiltration of inflammatory cells. Ct group (white), UC group (black), UC/preBact A group (vertical stripes), and UC/preBact B group (horizontal stripes). Values are shown as the mean ± SE, n = 5/group. The data were verified by one-way ANOVA. (* *p* < 0.05 vs. Ct group).

**Figure 7 microorganisms-11-01858-f007:**
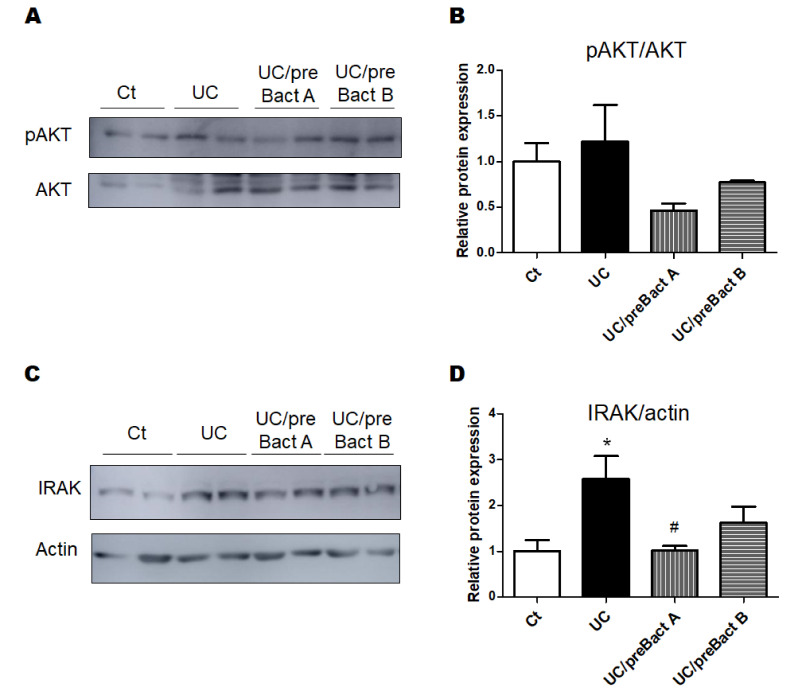
The protein expression at the colon in the preventive treatment mice with Bact A and Bact B. (**A**) The image of phosphor-AKT and AKT expression. (**B**) The protein expression of phosphor-AKT was quantified and normalized to that of AKT by Western blot. Ct group (white), UC group (black), UC/preBact A group (vertical stripes), and UC/preBact B group (horizontal stripes). Values are shown as the mean ± SE, n = 5/group. The data were verified by one-way ANOVA. (**C**) The image of IRAK and actin expression. (**D**) The protein expression of IRAK was quantified and normalized to that of actin by Western blot. Ct group (white), UC group (black), UC/preBact A group (vertical stripes), and UC/preBact B group (horizontal stripes). Values are shown as the mean ± SE, n = 5/group. The data were verified by one-way ANOVA. (* *p* < 0.05 vs. Ct group, ^#^
*p* < 0.05 vs. UC group). IRAK, interleukin-1 receptor-associated kinase.

**Table 1 microorganisms-11-01858-t001:** Disease activity index.

Score	Weight Loss (%)	Occult Blood	Stool Consistency
0	<1	Negative	Normal
1	1–5	Occult blood stool	Soft stool
2	5–10	Bloody stool	Loose stool
3	10–20		
4	>20	Hematochezia	Diarrhea

**Table 2 microorganisms-11-01858-t002:** Histological score.

Score	Epithelium Loss (%)	Crypt Damage (%)	Infiltration of Inflammatory Cells
0	None	None	None
1	0–5	0–5	Mild
2	5–10	5–10	Moderate
3	>10	>10	Severe

## Data Availability

Not applicable.

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
