# Peer review of "Gut Protective Effect from Newly Isolated Bacteria as Probiotics against Dextran Sulfate Sodium and Carrageenan-Induced Ulcerative Colitis"

_microorganisms, 2023, doi:10.3390/microorganisms11071858_

Round 1

Author Response

Reviewer1

Review

Gut protective effect from certain probiotics against DSS and carrageenan-induced ulcerative colitis

The authors have investigated potential gut protective effect of two unknown bacteria (patents pending) in chemically induced colitis in ICR mice. The novelty and impact of the findings are a bit hampered by not knowing which bacteria was used. Several other probiotic bacteria have been tested in similar mouse models previously. If possible, impact will be a lot higher if waiting for the patent to be approved. That said, the findings are of interest to the scientific community as it brings attention to the potential of using probiotic bacteria as a possible treatment principle in IBD.

Major comment:

  • If fecal samples were collected, it would strengthen the study to show if the intervention induced general microbiota alterations, such as F/B ratio etc.

That is a very good idea. As the samples have been lost, we would execute this in the forthcoming experiments.

  • English language should be thoroughly revised. First part of the introduction as well as the discussion are nicely written, but in general, there are a lot of unfinished sentences, unnecessary wrongly used words.

According to the suggestion, we have gone over the text/abstract and amended typos and grammatical errors as much as possible with a help of native English-speaker to improve the manuscript more helpful to the readers.

Examples:

  • Abstract: Do you mean “commensal” instead of “confident” bacteria?

Thank you so much. Yes, it should be “commensal”.

  • Intro second paragraph: “comparable symptoms could be brought in mice----“. Would rather write could be induced.

Amended.

  • Intro second parapraph: “with this UC model mice” (rather write UC mouse model). And this sentence is a repetition from another sentence above.

The sentence has been improved, accordingly.

  • Line 47,48. Bad English sentence. Rather “Gut microbial dysbiosis may also result from alterations in commensal microbiota composition”

Improved. Thank you so much.

  • Line 49: remove “the”

Removed.

  • Line 50: probiotics could work as a good mediator

Corrected.

  • Line 52: Replace “improved” by “dampened”

Replaced. Thank you.

  • Line 54-56: Repetition from earlier

This sentence has been altered, accordingly.

  • Line 57: uncomplete sentence. Do you mean “In addition to probiotics being potential inhibitors of pathogenic growth, certain probiotics----"

That is fine. Thank you.

  • Line 59: “to keep the integrity of mucosa in gut”. Should be “keep/maintain gut mucosal integrity”.

That is also fine. Thank you again.

  • Line 72: “it has been struggled to look for”. Rather, “it has been challenging to identify bacteria---”

Thank you so much. The sentence has been amended, so.

  • Line 75: Sufficient to write “unpublished data”

Amended, accordingly.

  • Line 78: Remove “in the present study”.

Deleted.

  • Line 78: “In conclusion after all”, this sentence does not fit here and should be in the conclusion.

This sentence has been transferred to the conclusion section. Thank you.

Minor comments:

  • How many mice were housed per cage?

One mouse per cage, which has been mentioned in the Materials and Methods section.

  • Body weight: I would prefer if you presented this as % (or % gain), as mice could have slight different reference weights at experiment start.

According to this suggestion, Figure2A and Figure 5A have been improved.

  • I cannot see any Table 1, regarding the disease activity score.

We have added Table 1 and Table 2 in the Materials and Methods section. Originally, Table 1 and Table 2 were provided in the manuscript, however, these were somehow lost in the present text.

  • What are the reasons for picking DSS/carr concentrations in the UC water, and duration? Has dose response been tested in these mice? I find it a bit strange that the mice do not loose any weight even though disease activity seems high in the UC group.

The DSS/carr concentrations and duration used in the present experiments for the development of UC had been determined from the consequence of preliminary experiments and the related literatures, which has been additionally described in the Result section. We think the stimulation of the DSS/carr concentrations may be mild for considerably losing body weight, but enough for making mucosal bleeding and/or diarrhea.

Reviewer 2 Report

REVIEW

Dear authors,

Please consider the following comments to improve the content of your manuscript before publication. 

MAJOR CONCERN:

-        The word "certain" is very general and does not specify whether they are commercial probiotic strains or recently isolated, they also state in the title that the strains used are probiotic when using this term, I recommend considering these comments for the better and adjusting the title to what that was carried out in the experimental work.

-        In section 2.1 Materials they seriously omit to describe in a general way the characteristics of the 2 bacteria used in the work (Bact A and Bact B), I understand that their microorganisms are in the process of patenting but they must describe at least their microscopic morphology, Gram, if they are lactic acid bacteria, and the source where they isolated these strains, which fruit?

-        In section 2.2 Mice, more information is missing on the administration of Bact A and Bact B, how much CFU did they inoculate with the bacteria in the UC water? DSS and k-carrageenam do not affect the viability of bacteria? Did you do the viability test before using this administration scheme? How often did they change the UC water with the bacteria? Do you know the volume of water the mice consumed per day and the amount of CFU they ingested?

-        This type of test where the administration of the bacteria is done in drinkers is not recommended. Peroral administration through a probe ensures that the amount of CFU of bacteria is inoculated and reaches the gastrointestinal tract. 

-        In section 2.3 Disease activity index score (DAI) Table 1 is not found.

-        In section 2.4 Histological examination Table 2 is not found.

-        Why did they not evaluate PI3K in Experiment 2 as in Experiment 1?

-        They mention in the discussion the anti-inflammatory capacity of the bacteria used, I recommend carrying out an evaluation of inflammatory cytokines in serum that complements the Western blot of PI3K/AKT, since if they actually have this immunomodulatory capacity they should significantly regulate the process systemic inflammation.

MINOR CONCERN:

-        The foot of Figures 23 and 4 are exactly the same as those of Figures 56 and 7, my recommendation is that to Figures 5, 6 and 7 add a text indicating the preventive treatment with Bact A and Bact B to be able to differentiate the experimental schemes.

-        Figures 3A and 6A are pixelated and the size is too small to observe histopathological changes, and most importantly, crypt damage, loss of epithelium and inflammatory cell infiltrate are not indicated, and were evaluated and graphed the Score.

It is necessary to make the following corrections in the indicated lines:

Line 86: the size of the word “Mice” is larger than the text.

Lines 293, 294: the size of the words “the used bacteria” is larger than the text.

Line 294: italicize the term “in vitro”.

Line 297: The size of the word "bacteria" is larger than the text.

Lines 304, 306, 311, 329, 330, 333, 334, 337: the size of the reference numbers “27, 28, 31, 35, 36, 38, 39, 40” is larger than the rest of the text.

Line 328: the size of the word “these bacteria” is larger than the text.

References

Most of the references are not written as requested by the journal, please review the format in detail and correct.

Please amend the requested comments and submit the revision file.

Author Response

Reviewer2

Dear authors,

Please consider the following comments to improve the content of your manuscript before publication. 

Thank you for your important comments.

MAJOR CONCERN:

-        The word "certain" is very general and does not specify whether they are commercial probiotic strains or recently isolated, they also state in the title that the strains used are probiotic when using this term, I recommend considering these comments for the better and adjusting the title to what that was carried out in the experimental work.

According to this suggestion, the title has been improved. Thank you so much.

-        In section 2.1 Materials they seriously omit to describe in a general way the characteristics of the 2 bacteria used in the work (Bact A and Bact B), I understand that their microorganisms are in the process of patenting but they must describe at least their microscopic morphology, Gram, if they are lactic acid bacteria, and the source where they isolated these strains, which fruit?

As mentioned in the discussion section, Bact A and Bact B are not lactic acid bacteria sp. All the microscopic morphology, Gram stain, and colony appearance have been included in the manuscript submitted elsewhere. Bact A and Bact B derive from strawberry and banana, respectively, which has been described in the Introduction section.

-        In section 2.2 Mice, more information is missing on the administration of Bact A and Bact B, how much CFU did they inoculate with the bacteria in the UC water? DSS and k-carrageenam do not affect the viability of bacteria? Did you do the viability test before using this administration scheme? How often did they change the UC water with the bacteria? Do you know the volume of water the mice consumed per day and the amount of CFU they ingested?

Prior to the experiments, we found that the growth of both bacteria at RT would make a platou in these conditon of DSS/ k-carrageenam drinking water. Hence, both bacterial concentrations for the peroral administration might be approximately 1 × 106 CFU ml-1 during the usage period.

-        This type of test where the administration of the bacteria is done in drinkers is not recommended. Peroral administration through a probe ensures that the amount of CFU of bacteria is inoculated and reaches the gastrointestinal tract. 

Both bacterial concentrations for the peroral administration were approximately 1 × 106 CFU ml-1, which has been described in materials and methods section. Your suggestion is absolutely correct. In the preliminary experiments of ours, both bacteria showed the heat, acid, alkali, and salts stability.

-        In section 2.3 Disease activity index score (DAI) Table 1 is not found.

We have added Table 1 in the Materials and Methods section. Originally, Table 1 was provided in the manuscript, however, this was somehow lost in the present text.

-        In section 2.4 Histological examination Table 2 is not found.

 We have also added Table 2 in the Materials and Methods section. Originally, Table 2 was also provided in the manuscript, however, this was somehow lost in the present text.

-        Why did they not evaluate PI3K in Experiment 2 as in Experiment 1?

It was all our fault. The first western data of PI3K in exp 2 was useless. Then, samples were gone, unfortunately.

-        They mention in the discussion the anti-inflammatory capacity of the bacteria used, I recommend carrying out an evaluation of inflammatory cytokines in serum that complements the Western blot of PI3K/AKT, since if they actually have this immunomodulatory capacity they should significantly regulate the process systemic inflammation.

That is a very good point, however, we had no serum samples now. Therefore, we would execute this in the forthcoming experiments.

MINOR CONCERN:

-        The foot of Figures 23 and 4 are exactly the same as those of Figures 56 and 7, my recommendation is that to Figures 5, 6 and 7 add a text indicating the preventive treatment with Bact A and Bact B to be able to differentiate the experimental schemes.

Thank you so much. We have amended the legends, accordingly.

-        Figures 3A and 6A are pixelated and the size is too small to observe histopathological changes, and most importantly, crypt damage, loss of epithelium and inflammatory cell infiltrate are not indicated, and were evaluated and graphed the Score.

 Figures 3A, 6A and their legends have been improved, accordingly.

It is necessary to make the following corrections in the indicated lines:

Thank you so much.

Line 86: the size of the word “Mice” is larger than the text.

Corrected.

Lines 293, 294: the size of the words “the used bacteria” is larger than the text.

Corrected.

Line 294: italicize the term “in vitro”.

Corrected.

Line 297: The size of the word "bacteria" is larger than the text.

Corrected.

Lines 304, 306, 311, 329, 330, 333, 334, 337: the size of the reference numbers “27, 28, 31, 35, 36, 38, 39, 40” is larger than the rest of the text.

Corrected.

Line 328: the size of the word “these bacteria” is larger than the text.

Corrected.

References

Most of the references are not written as requested by the journal, please review the format in detail and correct.

All references have been corrected to the journal style requested. 

Round 2

Reviewer 1 Report

The authors have made sufficient corrections for the manuscript to be accepted after correcting the following: In figure legends regarding the DAI scoring, it says that the score sums the daily score of five grades of weight loss and stool consistency and four grades of occult blood. But wasn't it rather also four grades of stool consistency? 

Reviewer 2 Report

The authors adequately answered each of the questions made, as well as carried out the corrections in the text.